# The Yin/Yang Balance of Communication between Sensory Neurons and Macrophages in Traumatic Peripheral Neuropathic Pain

**DOI:** 10.3390/ijms232012389

**Published:** 2022-10-16

**Authors:** Roxana-Olimpia Gheorghe, Andreea Violeta Grosu, Melania Bica-Popi, Violeta Ristoiu

**Affiliations:** Department of Anatomy, Animal Physiology and Biophysics, Faculty of Biology, University of Bucharest, 91-95 Splaiul Independentei, District 5, 050095 Bucharest, Romania

**Keywords:** traumatic neuropathic pain, macrophages, nociceptors, cytokines, chemokines, miRNA

## Abstract

Traumatic peripheral neuropathic pain is a complex syndrome caused by a primary lesion or dysfunction of the peripheral nervous system. Secondary to the lesion, resident or infiltrating macrophages proliferate and initiate a cross-talk with the sensory neurons, at the level of peripheral nerves and sensory ganglia. The neuron–macrophage interaction, which starts very early after the lesion, is very important for promoting pain development and for initiating changes that will facilitate the chronicization of pain, but it also has the potential to facilitate the resolution of injury-induced changes and, consequently, promote the reduction of pain. This review is an overview of the unique characteristics of nerve-associated macrophages in the peripheral nerves and sensory ganglia and of the molecules and signaling pathways involved in the neuro-immune cross-talk after a traumatic lesion, with the final aim of better understanding how the balance between pro- and anti-nociceptive dialogue between neurons and macrophages may be modulated for new therapeutic approaches.

## 1. Introduction: Neuropathic Pain Associated with Traumatic Peripheral Nerve Injuries

Peripheral nerve fibers are prone to being damaged easily by compression, crush, or transection, which results in nerve injuries of different degrees of severity. Peripheral nerve injuries are considered major health issues, given their higher prevalence and their deleterious impact on the motor activity and on the sensations and perceptions from the skin and joints, which frequently results in lifelong disability of an individual [1]. Depending on the degree of demyelination, the extent of damage to the axons and to the connective tissue, peripheral nerve injuries have been classified into several types according to Seddon [2] and expanded by Sunderland [3] (Figure 1): **Neurapraxia** is the mildest type of nerve injury, caused by local ischemia, traction, mild crush or compression. It is defined by a temporary blockage of nerve conduction caused by a segmental demyelination. **Axonotmesis** is caused by a nerve crush (due to a bat, a surgical clamp, or other crushing object) that does not result in a transection. It is characterized by a loss of axonal continuity, but the connective tissue surrounding the nerve is preserved. **Neurotmesis** is the most severe. It is caused by a nerve crush (due to laceration from a knife, gunshot, glass, etc.), which determines a full transection of the axons and connective tissue layers, with a complete loss of axonal continuity. A more extensive characterization of these types of peripheral nerve injuries has been reviewed elsewhere [4,5].

In axonotmesis and neurotmesis, axon regeneration is the primary goal. For a full recovery, the nerve must undergo three main processes: Wallerian degeneration (WD) (the clearing process of the remaining debris), axonal regeneration and end-organ reinnervation. WD takes place within the first 2 weeks after injury and includes: (1) fragmentation and degeneration of the axons distal to the lesion and (2) phagocytosis of the myelin sheath by Schwann cells and macrophages with the purpose of removing the debris, cleaning the area and allowing the peripheral nerve to regrow/regenerate from the segment proximal to the injury [6]. As the events associated with WD progress, more and more resident or hematogenous macrophages are recruited to the lesion site, which emphasizes their critical role for the WD and subsequent regeneration [7]. However, their contribution comes with a price, and this price is pain.

The pain that develops after an extensive nerve injury is usually chronic (persists for 3 months after trauma) [8] and neuropathic in nature. Chronic pain represents a major cause of human suffering worldwide (~20% of adults in developed nations) [9,10,11,12] especially because effective, specific and safe therapies have yet to be developed. Many of the **changes** that occur in response to nerve injury are **adaptive**, promoting repairing and healing, such as: removal of cell and myelin debris, changes in ionic channels and receptors to balance the loss of input, recruitment of antiapoptotic survival mechanisms to prevent neuronal death, induction of axon growth and sprouting, synaptic modelling and remyelination [13]. There are, however, many **maladaptive changes,** such as: abnormal stimulus thresholds and sensitivity, ectopic impulse generation, conduction slowing or block, reduced inhibition, inappropriate connectivity, abortive growth, neuronal loss and glial scarring [14,15]. These maladaptive changes induce an increase in pain sensitivity along the pain pathway, both in the peripheral component of the pain pathway, in a process known as **peripheral sensitization**, and in the central component of the pain pathway, in a process known as **central sensitization** [16,17]. Peripheral sensitization is characterized by an increased responsiveness and reduced spiking threshold of peripheral nociceptive neurons to stimulation of their receptive fields, induced by the local release of **primary mediators** either from primary sensory neurons or from the immune cells [18]. Central sensitization is characterized by an increased responsiveness of nociceptive neurons in the central nervous system (CNS) to their normal or subthreshold afferent input, induced by the release of **secondary mediators** from primary afferent terminals in the spinal dorsal horn, or **tertiary mediators** from microglia or astrocytes in the spinal cord [18]. Peripheral and central sensitization are frequently accompanied by **allodynia** (painful response to non-noxious stimuli), **hyperalgesia** (exaggerated response to noxious stimuli), or **spontaneous pain** [14], which significantly reduce the quality of life of patients and require therapeutic interventions. To understand the pathogenesis of traumatic neuropathic pain, many rodent surrogate models have been developed, such as: sciatic nerve transection (SNT), partial sciatic nerve ligation (PSNL), spinal nerve ligation (SNL), spared nerve injury (SNI) and chronic constriction injury (CCI) [19,20]. Even though these models are not always very good predictors of the involvement of particular targets or processes in human neuropathic pain and are incomplete because the subjective component of pain cannot be evaluated, they have great utility in exploring the cellular and molecular processes responsible for pain.

Neuropathic pain is mainly the result of a **neuronal dysfunction** [14,15,17,21]. However, there is a growing body of evidence indicating that the cause of neuropathic pain is not restricted to changes in neuronal activity, but may involve numerous **neuro-immune interactions** among neurons, immune cells and immune-like glial cells in the periphery and CNS, mediated by inflammatory cytokines and chemokines [22,23,24,25] and facilitated by ion channels expressed in either neurons or immune cells. Among the immune cells involved in these neuro-immune interactions, macrophages, which are the primary sensors of danger in the host, seem to play an important role. In this review, we will outline the molecules that mediate the interactions between macrophages and sensory neurons from the peripheral nervous system and facilitate the development and maintenance of neuropathic pain, by integrating data mainly from animal models of traumatic peripheral neuropathic pain.

## 2. Macrophages in the Peripheral Nervous System: A Distinct Population

Macrophages are present in all tissues and have a chameleon-like phenotype, changing their physiology easily as a response to environmental stimuli [26]. They act as house-keeping cells, performing immune surveillance in order to remove, repair and replace cells or matrices lost through senescence or following wounding. They do these by processes such as phagocytosis and cytokine/growth factor secretion, which all belong to the innate immune response, or by antigen presentation through which the connection with the adaptive immune system is made [26]. At a given moment, depending on the signal from the environment, macrophages can exist in an M0 (resting) phenotype from which they switch to M1 (pro-inflammatory) state or M2 (anti-inflammatory) state, with many intermediary states in between [27,28].

In the peripheral nervous system, macrophages are localized in the nerves at the epineurium level, where they are typically described near blood vessels and in the endoneurial space where they represent 2–9% of all endoneurial cells, or in the sensory ganglia scattered between neuronal cell bodies [29]. In the last years, the transcriptional identity and functions of macrophages in the peripheral nervous system were intensively studied, and new information about their origin [30] and features [29,31] have been obtained.

**In the nerve**, studies of single-cell RNA sequencing have revealed the presence of two major subsets of macrophages inside the mouse sciatic nerve, with distinct locations: one located in the endoneurium (Relmα^−^Mgl1^−^Iba1^+^ snMac1 or R1 population) and one in the epineurium (Relmα^+^Mgl1^+^Iba1^+^ snMac2 or R2 population) [31]. Additional transcriptomic analysis has revealed that the endoneurial R1 population expresses genes specific to microglia (the equivalent of macrophages in the CNS), such as CX3CR1, CCL2, or Trem2, and that, compared to the epineurial R2 population, it expresses more immediate early response genes, such as Atf3, Fos or Jun [31]. These data suggest that the R1 population, which resides within the nerve fascicles and is in close contact with neurons and Schwann cells, is intrinsically more responsive. Differences have been observed not only inside a nerve, but also between nerves in mice: a transcriptomic study identified 24 genes enriched in sciatic nerve macrophages, 23 genes enriched in cutaneous intercostal (fascial) nerve macrophages, and 12 genes enriched in vagus nerve macrophages [29]. Interestingly, the upregulated genes in mouse sciatic nerve-associated macrophages resemble genes expressed in macrophages isolated from sympathetic nerve fibers associated with visceral fat and genes expressed by macrophages derived from cutaneous and visceral fat [32,33].

**In the dorsal root ganglia (DRG)** (more commonly investigated than other sensory ganglia), macrophages have been identified as belonging to the same two distinct CD11b (+)/Ly6G (−) myeloid subpopulations as in the nerve, that is MHCII (+) and MHCII (−)/Ly6C (−) [33], likely equivalent to CX3CR1 (+) resident populations described elsewhere [29]. However, in comparison with nerve-resident macrophages, macrophages residing within the mouse DRG were found to express 79 upregulated genes and 52 downregulated genes [29]. Interestingly, the upregulated genes in mouse DRG-associated macrophages resemble genes expressed in macrophages isolated from sympathetic ganglia and genes commonly expressed by resident microglia [32,33,34].

These data suggest that although macrophages in the peripheral nervous system (PNS) are transcriptionally similar, significant differences exist between those adjacent to axons and those residing close to neuronal cell bodies, which is conceivable given the fact that tissue-resident macrophage populations are distinct from one another, and that their development and phenotype are dependent on local environmental cues [35,36].

## 3. Macrophages Are Quickly Activated after a Traumatic Peripheral Nerve Lesion

After a traumatic injury at the peripheral nerve level in mice, MHCII (+) and MHCII (−)/Ly6C (−) resident macrophages are joined by Ly6C (+) and MHCII (+)/Ly6C (+) infiltrating macrophages [33], which accumulate at the injury site at the nerve level, but also around the neuronal cell bodies in the sensory ganglia, and switch to the M1 phenotype. **At the nerve level,** 1 and 5 days after *crush injury* on the sciatic nerve in mice, resident macrophages (epineurial and endoneurial) represented only 5–10% of the total population, while the rest (90–95%) were infiltrating macrophages, most of them appearing at the injury site 1 day after the lesion [31]. Among the resident macrophages, it has been shown that the epineurial macrophages did not significantly react to the nerve injury, while endoneurial macrophages reacted by producing monocyte chemoattractants [31]. In the first 5 days after crush injury in mice, the recruited macrophages adopted the resident PNS macrophage signature (i.e., lost Ly6C expression and gained CD64 expression) and remained in the PNS during the repairing process, such that 1 month after the injury, macrophages were still predominantly derived from recruited circulating precursors and did not revert to the original resident macrophage population [31]. In contrast, after a *partial sciatic nerve injury* in mice, both resident (MHCII (+) and MHCII (−)/Ly6C (−)) and infiltrating (Ly6C (+) and MHCII (+)/Ly6C (+)) macrophages were significantly upregulated 1 to 3 days and 2, 10, and 14 weeks after injury [33]. These data suggest that after a traumatic injury, although both resident and infiltrating macrophages are recruited to the nerve lesion site, how they will respond depends largely on the type of injury.

After a traumatic injury **at the sensory ganglia level**, in particular DRG, macrophages form ring-like clusters around neuronal bodies, as it has been described for several traumatic neuropathic pain models: 5 days after spinal nerve ligation [37], 7 days after spared nerve injury [38], 7 days after sciatic nerve transection [39] or 7 days after sciatic nerve ligation and transection [40]. However, macrophages do not always behave like that inside DRG, as it has also been previously reviewed [41]. For example, in rats with diabetic neuropathy, macrophages do not cluster around neuronal bodies, but remain scattered between neuronal cell bodies, although the number of macrophages is increased from the moment of induction of diabetes [37], which reconfirms again that macrophages’ responses are dependent on local environmental cues.

To separate resident from infiltrating macrophages at the DRG level, studies on MAFIA (macrophage Fas-induced apoptosis) transgenic mice proved very useful [42]. More specifically, MAFIA mice carry a suicide gene expressed in macrophages, which allows a targeted, specific killing of peripheral macrophages by a specific compound, AP20187, which, unlike other drugs, fails to cross the blood–brain barrier. Mice in which a specific depletion of resident macrophages from DRG was thus performed developed mechanical allodynia 7 days after SNI, compared with 24 h later in non-treated animals, and interestingly, showed no sign of hematogenous macrophages infiltration in the meantime [42]. These data suggest that, at least for this neuropathic pain model and in contrast with the response at the nerve level, resident macrophages rather than infiltrating ones seem to be more involved in the first days after the lesion.

In the first hours after a traumatic peripheral nerve lesion, **at the nerve lesion site**, Schwann cells, fibroblasts and endoneurial macrophages activate, releasing chemoattractant factors which will recruit hematogenous macrophages starting 2–3 days after the lesion, with a peak at about 7 days [6,31]. In the first hours (5–10 h) after a nerve lesion, Schwann cells secrete pro-inflammatory cytokines (i.e., TNFα, IL-1α, IL-1β), which will induce in an autocrine manner their own secretion of chemoattractant chemokines (i.e., CCL2, CCL3) [43,44,45] and in a paracrine manner, in nearby resident fibroblasts, the secretion of IL-6 and GM-CSF, which downstream will act as an algesic agent and inducer of macrophages’ polarization towards the M1 phenotype (see below) [6] (Figure 2). Independently or activated by the cytokines released by Schwann cells (not clear yet), endoneurial macrophages start to release monocyte-macrophage chemoattractants (i.e., CCL6, CCL7, CCL8, and CCL12) on day 1 after injury [31] (Figure 2). Recruitment is further aided by TNFα-dependent induction of the matrix metalloproteinase MMP-9 that Schwann cells produce [46] and by complement [47].

At the **sensory ganglia level**, in particular DRG, macrophages are attracted around DRG neurons by different chemoattractant molecules.

The **CCL2/MCP1** chemokine (Figure 3) is upregulated in DRG neurons after SNL [48], SNI [49], PSNL [50], sciatic nerve ligation and transection [40] or CCI [51], and facilitates clustering of macrophages around DRG neurons since this clustering does not happen anymore in CCL2^−^/^−^ animals [40]. The same clustering effect was obtained by just overexpressing CCL2 in mouse L5 DRG using viral transfection methods [52]. However, CCR2 (+) (the CCL2 receptor) macrophages have been also identified in L4/L5 DRG of uninjured CSF1R-GFP^+^/^−^ & CCR2-RFP^+^/^−^ mice, suggesting that there might be an endogenous secretion of CCL2 to which resident macrophages can respond [42].

**CSF1/M-CSF** (Figure 3) is a cytokine secreted at the mRNA level by DRG neurons after SNI as soon as 2 days after the lesion [53], which stays up 3–7 days after SNI in mouse DRG neurons specifically surrounded by rings of GFP (+) macrophages [42]. Conditional deletion of the CSF1 gene abrogated macrophage proliferation and clustering at PO4 after SNI in the ipsilateral axotomized DRG, but these events were not compromised in mice globally lacking CCL2, which suggest that the macrophages’ attraction around DRG neurons is a CSF1-dependent and CCL2-independent process [42]. Considering that the inhibition of the CSF1 receptor also prevented the proliferation of resident macrophages in rats [54], it can be concluded that the ring of the macrophages formed around DRG neurons is the result of a combination of chemotaxis and local proliferation CSF1-dependent, at least after SNI.

In addition, some other chemokines might also contribute to the ring formation, although additional experimental data are still necessary (Figure 3). Thus, injections of **CCL3** and **CX3CL1** directly into the L5 rat DRG leads to a macrophage accumulation in the ganglion comparable to that which occurred after sciatic nerve lesion or after injection of CCL2 [40]. In addition, after L5 spinal nerve transection injury in mice, CCL3, whose expression increases at significant levels at 8 h after the lesion, to constantly decrease at 24 and 48 h later, is, at least partly and in a TLR2-dependent manner, responsible for the macrophage infiltration in the DRG [55]. On the other hand, CX3CL1 is expressed in all rat DRG neurons regardless of size in control conditions, but no increase was noticed after CCI [56], similar to the sciatic nerve, where no increase was found after axotomy in mice [57]. These data put CX3CL1′s contribution to the ring formation under a question mark, although its role in neuropathic pain development at the level of the spinal cord is very well documented [58].

In an RNA-seq study on mice, increased levels for mRNA for **CCL4, CCL7, CCL9,** and **CCL12** in the DRG were reported after injury, but the effects of these molecules on macrophage infiltration and a possible ring formation around DRG neurons in vivo remain to be tested [59] (Figure 3). These chemokines may also act as **secondary mediators** between DRG neurons and the dorsal horn of the spinal cord, where they will activate spinal microglia, causing them to further release **tertiary mediators** which will activate spinal neurons [58]. In this way, spinal microglia can detect and respond to peripheral nerve injury and participate in the central sensitization processes mentioned above. The following chemokines are well documented as secondary mediators: (1) CSF1: it can be released from primary afferent terminals and activates microglia [53]; (2) CCL21: vesicles containing CCL21 are preferentially transported into axons [60], and it can be released from terminals of injured neurons [61] and affects microglia functioning [62]; (3) CCL2: it is known that it is expressed in vesicles in DRG neurons, and it is released in a Ca^2+^-dependent manner [63], but it was not identified in the primary afferent terminals ending in the dorsal horn [64]. Therefore, rather than functioning as a secondary mediator between primary afferents and spinal microglia, CCL2 may fulfill an autocrine or paracrine function within the DRG.

## 4. There Is an Intense Cross-Talk between Macrophages and Sensory Neurons Facilitated by Pro-Inflammatory Mediators, in Which Ion Channels Are Important Players

As mentioned above, macrophages that accumulate at the injury site at the nerve level or around the neuronal cell bodies in the sensory ganglia switch to the M1 phenotype. This switch is facilitated by some of the chemoattractant molecules that may have more than a chemotactic role. Thus, direct stimulation of macrophages with **CCL2** favors M1 polarization, causing upregulation of TNFα in the macrophage cell line RAW264.7 [65], increased TNFα expression in murine peritoneal macrophages [66], upregulation of IL-1β, iNOS, and IL-6 mRNA levels in murine alveolar macrophages and BMDMs [67], and activation of NOX2-dependent oxidative burst in macrophages infiltrated in the mouse sciatic nerve after PSNL, further on associated with mechanical allodynia development via TRPA1 activation in Schwann cells and DRG neurons [43]. In contrast, on human macrophages, CCL2 seemed to rather facilitate the acquisition of an M2, anti-inflammatory phenotype, by modulating the expression of functionally relevant and polarization-associated genes [68]. **GM-CSF** secreted at the peripheral nerve level by the fibroblasts facilitated the M1 polarization as well [69], while **CSF1**, a powerful chemoattractant at the DRG level seemed to be, surprisingly, associated with M2 polarization [70]. **TNF-α,** a pro-inflammatory cytokine, was shown to also stimulate M1 polarization [71], possibly facilitated by the iron accumulation in macrophages, as it was shown in the case of macrophages infiltrated in the spinal cord after spinal cord injury in mice [72].

The close vicinity between M1 macrophages and sensory neurons facilitates an intense bidirectional communication via different **primary mediators**, which contributes to the establishment of a peripheral sensitization process as part of the traumatic neuropathic pain development process [73,74].

Many of the primary mediators **released by macrophages** have as a final target neuronal ion channels, whose altered properties contribute to increased neuronal excitability.

The **IL-1β** and **TNFα** cytokines, whose implications in neuropathic pain pathogenesis are extensively documented [75], act directly on primary sensory neurons and substantially increase their excitability by augmenting TTX-resistant sodium current via p38 MAP kinase in rats [76] (Figure 4).

The **IL-6** cytokine, strongly related to neuropathic pain pathogenesis [77] and specifically associated with macrophages after PSNL injury [78], contributes to an increased neuronal excitability by the upregulation of Ca_V_3.2 T-type channel expression and function through the IL-6/sIL-6R trans-signaling pathway via a JAK/STAT signaling in rats [79] (Figure 4).

**PGE2**, a pro-inflammatory prostanoid specifically associated with macrophages and traumatic neuropathic pain [80,81], causes pain by direct or indirect actions on nociceptors. *Directly*, PGE2 acting on specific receptors and via PKC and PKA signaling pathways sensitizes the TRPV1 receptor in mice [82] and bradykinin receptors in rats [83], facilitating a faster response of nociceptors to subsequent molecules and, therefore, the generation of an increased number of action potentials. *Indirectly*, PGE2 stimulates the release of pain-related neuropeptides, such as: (1) SP and CGRP via PKC and PKA signaling pathways [84]; (2) CCL2 chemokine via a not-clear-yet signaling pathway, which is downstream of the EP3 receptor [85]; (3) IL-6 cytokine via EP4/PKA/PKC/ERK signaling pathways [80]; and (4) BDNF via EP1&EP4/PKA/CREB and EP1&EP4/ERK/CREB signaling pathways [80]. Macrophages, which do not express BDNF, are nevertheless required for the nerve injury-induced upregulation of BDNF in mouse axotomized sensory neurons [42] (Figure 4).

**CCL2** is secreted not only by sensory neurons, but also by macrophages [86]. Directly, via CCR2, CCL2 has excitatory effects on rat primary sensory neurons in a chronic compression of the DRG (CCD) neuropathic pain model [87], due to the activation of a non-voltage-dependent depolarizing current with characteristics similar to a non-selective cation conductance, possibly TRP channels [63], and inhibition of a voltage-dependent outward current [88] (Figure 4).

**CXCL1** is another chemokine that significantly increases in rats after traumatic peripheral nerve injury [89], but its origin under these circumstances (i.e., either satellite cells or macrophages) was not clearly established [90]. When released locally, via CXCR2, CXCL1 amplifies the peripheral neuronal excitability by increasing Na_v_ 1.1 and 1.7 currents in both IB4 (+) (nonpeptidergic sensory neurons) and IB4 (−) (peptidergic sensory neurons) DRG neurons without altering their voltage dependence or channel-opening kinetics [91]. This effect was further on correlated with a marked increase in excitability (i.e., resting potential depolarization, decreased rheobase, lower action potential threshold) and a striking ability of neurons to fire repetitively (Figure 4). In addition, opposite effects were noticed on IB4 (+) and IB4 (−) neurons upon CXCL1 stimulation, even though CXCR2 are expressed on both types of neurons. More specifically, it was shown an increase in the transient and sustained potassium currents only in IB4 (−) neurons after overnight incubation, which may facilitate repetitive firing [92], while the voltage-activated K+ current of IB4 (+) neurons was unchanged (Figure 4).

In 2010, it was reported that the high-mobility group box 1 (**HMGB1**) protein was upregulated in rat primary afferent neurons and satellite glial cells in the DRG and in the Schwann cells of the spinal nerve after SNL, providing the first evidence for its role in neuropathic pain pathogenesis [93]. Later on, increased HMGB1 expression was detected also in mice, in both infiltrating macrophages and proliferating Schwann cells in the sciatic nerve 14 days following PSNL [94], confirming HMGB1 as a new mediator which facilitates the communication between macrophages and sensory neurons. Interestingly, mechanical hyperalgesia induced by HMGB1 intraplantar administration in mice proved to be dependent of RAGE and TLR4 [95], which downstream induced early growth response 1 (Egr-1) transcription factor upregulation via the RAGE pathway, and further down the Ca_V_3.2 overexpression in L4 DRG after L5 spinal nerve cutting in rats [96] (Figure 4).

Reactive oxygen species (**ROS**) released by macrophages were specifically related to TRPA1-mediated mechanical allodynia and neuroinflammation in a PSNL neuropathic pain model performed on mice [43]. The authors proposed a comprehensive mechanism in which ROS released by either macrophages or nociceptors maintain, in a spatially confined manner, macrophage infiltration into the injured nerve and send paracrine signals to activate TRPA1 of Schwann cells surrounding the nociceptors, to sustain mechanical allodynia [43]. More specifically, 10 days after surgery, CCL2 released at the site of the injury would promote the extravasation of hematogenous monocytes/macrophages, which would generate a rapid burst of ROS (via the NOX2-dependent oxidative pathway), which would further activate the TRPA1 channel localized in Schwann cells. TRPA1 activation in Schwann cells will evoke a Ca^2+^-dependent, NOX1-mediated prolonged H_2_O_2_ generation with a dual function: the outward H_2_O_2_ release would produce a space-scaled gradient that would determine the final macrophage influx to the injured nerve trunk, whereas the inward H_2_O_2_ release would target the nociceptor TRPA1 to produce mechanical allodynia [43] (Figure 4). Although not specifically related to a traumatic neuropathic pain model, it was shown that ROS released by activated macrophages upon intraplantar administration of angiotensin II further on activated TRPA1 on mouse and human DRG sensory neurons via cysteine modification of the channel [97]. Interestingly, intrathecal administration of angiotensin II in mice did not induce mechanical hypersensitivity.

**Sensory neurons** respond to the signals received from macrophages, not only by increasing their excitability, but also by **secreting other primary mediators**, which, in a positive feedback loop, act on macrophages, further activating them, thus reinforcing peripheral sensitization and pain development. Ion channels in the macrophages can be the target of this neuron–macrophage back communication.

As mentioned above, **CCL2** released by injured neurons not only acts as a chemoattractant molecule, but also promotes the activation of NOX2-dependent oxidative burst in macrophages and further on ROS-dependent TRPA1 activation in Schwann cells and DRG neurons [43]. The CCL2-dependent recruitment of macrophages to the nerve injury site is most likely initiated by the mechanically activated cationic channel Piezo1, as it was suggested for macrophages involved in renal fibrosis [98]. In their study, the authors showed that Piezo1 gene deletion in the myeloid lineage of genetically engineered mice prevented the development of renal fibrosis by decreasing macrophage recruitment to the injured kidney, most likely by regulating CCL2–CCR2 signaling via Notch activity, and by restraining macrophage activation and secretion of pro-inflammatory cytokines (IL-1β, TNF-α, IL-6) via a calpain-dependent signaling pathway (Figure 4). After a traumatic nerve injury, the mechanical properties of the nerve and of the surrounding tissues, including stiffness, architecture, and molecular and cellular composition, change dramatically. There are no clear data whether Piezo1, putatively expressed by nerve-associated macrophages too, upon activation by the nerve injury-induced mechanical stress, is involved in initiating the NOX2-dependent oxidative burst in macrophages after PSNL as mentioned above or whether other signaling pathways are activated downstream. However, since physical factors strongly coregulate macrophages’ plasticity and phagocytosis [99] and Piezo1 seems to be very important in modulating macrophages’ polarization and stiffness sensing [100], it is highly likely that this ion channel is strongly involved in the dialogue macrophages initiate with neurons after a nerve injury. As also mentioned above, PGE2 secreted by injury-activated macrophages may stimulate in DRG neurons the release of CCL2, and thus, the CCL2-mediated communication loop between neurons and macrophages is complete.

Upon PGE2 activation, DRG neurons may secrete other primary mediators as well, some of which further increase the positive feedback loop and increase neuronal excitability, others facilitate the M1 to M2 switch of macrophages and initiate an anti-nociceptive dialogue between macrophages and DRG neurons, and others may act as secondary mediators and facilitate communication further up at the spinal cord level. Substance P, CGRP, IL-6, and BDNF induced by PGE2 may have one or more of these roles.

**Substance P (SP)** was shown to enhance the secretion of the inflammatory chemokines MIP-2 and CCL2 by murine macrophages via ERK/p38 MAPK-mediated NF-kappaB activation [101] or of the TNF-α pro-inflammatory cytokine via the JNK and p38 MAPK pathway [102], all specifics to the M1 pro-inflammatory macrophages (Figure 4). Interestingly, SP can also directly induce M2 polarization and IL-10 secretion in rat macrophages, even in the presence of IFNγ, a potent M1-skewing cytokine [103] (Figure 5). This switch is possible because macrophages have a unique, flexible metabolism, able to generate customized responses, even antagonistic ones to the same metabolite, like in the case of substance P. This may explain how, during WD, at around 7 days after a nerve injury, macrophages start to switch towards M2 anti-inflammatory phenotype and begin to secrete anti-inflammatory cytokines, such as IL-10, which will help in the resolution of inflammation and pain. In parallel to this process, axons’ regeneration is promoted by Schwann cells, which form Büngner bands to guide the regrowing of axons from the proximal end. Therefore, even though they may be “bathed” in a pro-inflammatory environment, macrophages have the potential to gradually switch to a healing phenotype due to their unique metabolism, in a very delicate Yin/Yang balance.

**CGRP** was strongly related to neuropathic pain, where it was shown to support both peripheral and central sensitization, without eliciting nociceptive signals alone to [104]. After a nerve injury, at DRG the level, CGRP may act in an autocrine manner and enhance TTX-resistant Na+ currents in small- and medium-diameter cultured DRG neurons [105] (Figure 4), while at the spinal cord level, upon release from the primary afferent terminals, it was shown to be able to activate signaling cascades that sensitize the NMDA receptor [106]. Interestingly, CGRP may act on macrophages as well, promoting the M2 phenotype in two ways: (1) by acting as a costimulus in parallel with other TLR4 agonists (i.e., LPS) to enhance the secretion of the anti-inflammatory cytokine IL-10 via the CGRP receptor and a PKA signaling pathway associated with a prolonged activation of CREB [107] (Figure 5), and (2) by stimulating phagocytosis [108]. These M2 pathways were not specifically associated with a traumatic pain model. Instead, after PSNL, M1-specific effects were described: increased levels of CGRP in injured neuroma and invading macrophages were associated with the upregulation of IL-6 in rat macrophages and with the maintenance of neuropathic pain [109] (Figure 4). Therefore, although not proven for traumatic neuropathic pain, neurons may have the potential to not only activate macrophages via CGRP, but also to induce their M2 switch and initiate the healing and pain resolution.

**IL-6** secreted by the DRG neurons upon PGE2 activation may act in an autocrine manner to increase neuronal excitability at the DRG level, but it may also act as a secondary mediator serving as an “off signal” to ensure the transient nature of injury-induced neuroinflammation by promoting a desensitized phenotype of microglia at the spinal cord level [110] (Figure 4). **BDNF** secreted by the DRG neurons upon PGE2 activation acts as a secondary mediator at the level of the spinal cord, mediating the transition from acute to chronic pain and contributing to the establishment of a central sensitization process as part of the traumatic neuropathic pain development process [111,112] (Figure 4).

## 5. In Parallel with the “Pro-nociceptive” Dialogue between Primary Sensory Neurons and Macrophages, There Is Also an “Anti-nociceptive” Dialogue with M2 Macrophages That Helps to Reduce Pain

As mentioned in Section 4, SP and CGRP have the potential to facilitate the M2 switch of macrophages while also sustaining pro-inflammatory activities, as a sort of primary buds of anti-nociceptive communication between neurons and macrophages. In addition to that, other substances that promote the M2 switch of macrophages and have an anti-nociceptive effect have been described. Their release confirms that, in fact, macrophages located closer to neurons are there to heal them after injury, but until this happens, the neurons have to pay a toll in increased excitability and pain.

It was recently shown that 7 to 14 days after SNI, mouse dorsal root ganglia neurons release increased amounts of **IL-27**, which, via the WSX-1 receptor expressed mainly in macrophages, stimulates macrophages to release at their turn increased amounts of the anti-inflammatory cytokine IL-10 [113]. SNI-induced neuropathic pain was enhanced in IL-27-deficient mice, whereas nociceptive pain was similar to that of wild-type mice, indicating that IL-27 signaling is important for controlling the development of neuropathic pain, but it is not involved in the detection of mechanical, thermal, and chemical nociception in basal conditions. Interestingly, IL-10 production was not abrogated in IL-27-deficient mice, suggesting that other mechanisms might be involved in the induction of IL-10, such as SP and CGRP mentioned in Section 4. The anti-nociceptive effect of IL-10 is via IL-10 receptors, which were shown to be expressed on rat primary sensory neurons, and is due to a reduction of TTX-sensitive and Na_v_1.8 currents and to a reverse of the increased sodium currents induced by TNF-α [114] (Figure 5).

In addition, M2 macrophages may attenuate neuropathic pain by releasing **opioid peptides (Met-enkephalin, dynorphin and β-endorphin)**, which bind to opioid receptors on nociceptors. It was shown that perineural transplantation of M2 macrophages (induced in vitro by treatment with IL-4) resulted in the amelioration of CCI-induced mechanical hypersensitivity even if the injections were performed on days 14 and 15 after injury [115]. Compared with M0 and M1 macrophages, M2 macrophages contained and secreted significantly higher levels of opioids, and their anti-neuropathic pain effect was compromised by opioid receptor blocking. β-endorphin and enkephalins are anti-nociceptive peptides, linking µ (mu) and δ (delta) opioid receptors, while dynorphins can mediate not only anti-nociceptive effects, via κ (kappa) opioid receptors, but also pro-nociceptive effects via NMDA receptors [116]. Opioid receptors are G-protein-coupled receptors expressed in the central and peripheral nervous systems, in neuroendocrine and immune tissues and cells [117]. In DRG neurons, N-type calcium channels along with opioid receptors can be co-internalized following prolonged agonist exposure, which may further reduce neurotransmitter release and the transmission of pain signals to CNS [118] (Figure 5).

A special category of compounds, the specialized pro-resolving lipid mediators (SPMs), may also contribute to the anti-nociceptive environment in the DRG, not necessarily in the first days after the lesion, but in the later stages of injury-induced inflammation. Recent preclinical and clinical studies have described potent anti-nociceptive effects of SPMs in different pain pathologies, either inflammatory pain (i.e., carrageenan-elicited pain, complete Freud’s adjuvant-induced pain, osteoarthritis pain, temporomandibular joint inflammatory pain), traumatic neuropathic pain (i.e., CCI, spinal cord injury, chronic compression of the dorsal root ganglia, chronic post-thoracotomy pain, postoperative pain induced by tibial bone fracture), or chemotherapy-induced pain [119]. SPMs are synthesized from polyunsaturated fatty acids (PUFA) (i.e., lipoxins (LX) derive from arachidonic acid, an omega-6 PUFA, and protectins (PD), maresins (MaR) and resolvins (Rv) derive from omega-3 fatty acids) and can be secreted by different types of cells, such as endothelial cells, leukocytes, monocytes, T cells and macrophages [119]. Interestingly, even though macrophages may secrete SPMs, they also express specific receptors for SPMs, which, either in an autocrine manner or directly if SPMs are coming from other source, may facilitate M2 polarization of macrophages and enhance their phagocytic activity to resolve inflammation [120,121]. Specifically for the traumatic neuropathic pain models discussed in this review, intrathecal administration of resolvin E1 and aspirin-triggered lipoxin A4 was associated with reduced CCI-neuropathic pain mainly by reducing microgliosis in the rat spinal cord, either as a pre-treatment, at 7 days after CCI or at 3 weeks after CCI [122,123]. However, local delivery of neuroprotectin D1/protectin D1 (NPD1/PD1) to the injury site of the sciatic nerve, at the time of surgery, was able to also completely prevent CCI-induced mechanical allodynia for 4 weeks and suppress the injury-induced infiltration of inflammatory macrophages inside DRG [124]. The analgesic effect of SPMs might also be due to blocking TRP channels on DRG neurons, via specific GPCR receptors: RvE1 inhibits TRPV1 via ChemR23 co-expressed in small DRG neurons [125], while RvD1 and D2 inhibit TRPV1 and TRPA1 via GPR18 (at least for RvD2) [126,127] (Figure 5). It is not very clear how much of the SPMs secreted at the lesion site or inside a DRG after a traumatic nerve lesion are coming from macrophages or from other cells, but the anti-nociceptive results are compelling, and more data about their role in neuron–macrophage communication are required.

## 6. MicroRNAs May Also Facilitate Pro- and Anti-nociceptive Effects on Primary Sensory Neurons, with Macrophage Involvement More or Less Well Documented

In the last years, **miRNAs**, small noncoding RNAs of 19–25 nucleotides that bind the 3‘ UTRs of mRNAs to regulate gene expression post-transcriptionally, have started to be considered potential new targets for pain therapy, given their implication in neuropathic pain development and/or maintenance both in humans and in animal neuropathic pain models, as comprehensively reviewed by Kalpachidou et al. [128]. Generated inside the cells (i.e., nucleus or cytosol), miRNAs are also detectable in exosomes, the extracellular vesicles that both immune cells and neurons can shed in their extracellular environment and in the body fluids (i.e., CSF, blood plasma or saliva), where they can be exploited for diagnostic purposes. In this review, we will concentrate on the pro- or anti-nociceptive effects of those miRNAs, which were specifically associated with traumatic neuropathic pain in PNS.

**miR-let-7b** was shown to be an important pain mediator in the peripheral nervous system, given the fact that it is able to induce, in a sequence-dependent manner requiring GUUGUGU motif, rapid inward currents and action potentials only in DRG nociceptors co-expressing TLR7 and TRPA1, and that intraplantar injection of miR-let-7b elicited **persistent mechanical allodynia**, which was abolished in mice lacking TLR7 or TRPA1 [129]. Notably, the authors showed that miR-let-7b is highly expressed in the DRG tissue and can be released from DRG neurons in an activity-dependent manner. Knowing that miR-let-7b can activate murine microglia and macrophages through TLR7 [130] and that it can also be released by human macrophages [131], it is highly possible that this miRNA may facilitate the cross-talk between macrophages and neurons after traumatic injuries, even though, so far, it was not associated with a particular traumatic neuropathic pain model (Figure 6).

In contrast, the contribution of **miR-21** to traumatic neuropathic pain is much better documented. Thus, 7 days after SNI, it was shown that the expression of miR-21 was upregulated in mouse DRG neurons with **pro-nociceptive effects**: the intrathecal delivery of a miR-21 antagomir and the conditional deletion of miR-21 in sensory neurons reduced neuropathic hypersensitivity and macrophage infiltration in the DRG [132]. The authors also confirmed the action mechanism: upon noxious stimulation, TRPV1 receptors are activated, which facilitates the release of miR-21 packaged in exosomes by DRG neuronal cell bodies, which are next readily engulfed by macrophages. Inside macrophages, miR-21 induces polarization towards a pro-inflammatory, pro-nociceptive M1 phenotype, which will further on facilitate pain signaling following peripheral nerve injury (Figure 6). An increase in miR-21 inside all mouse DRG neurons was also confirmed 10 days after SNL, where it was proposed that it acts in an autocrine manner via TLR8 receptors only in small and medium-sized neurons, which are the only types of DRG neurons that coexpress miR-21 and TLR8 receptors [133]. After SNL injury, miR-21 could be released by large neurons as well, and be taken up by small and medium neurons to reach endosomal TLR8. TLR8 is not expressed in the membrane of DRG neurons, indicating that it cannot directly control plasma membrane excitability. Instead, TLR8 is expressed in the subcellular endoplasmic reticulum, endosomes and lysosomes of DRG neurons, similar to its location in monocytes and macrophages [133]. Intrathecal injection of miR-21 decreases the paw withdrawal threshold, increases p-ERK expression in the DRG in WT mice but not in TLR8^−^/^−^ mice, and further increases the number of action potentials in the whole-mount DRGs after incubation with miR-21 in the DRG neurons of WT mice but not TLR8^−^/^−^ mice. Even though ionic channel targets are not clear in this case, all these data confirm that miR-21 is involved in neuropathic pain pathogenesis (Figure 6).

A similar polarizing effect towards an M1 phenotype was shown for **miR-23a**, upregulated in mice DRG neurons after SNI [134]. Capsaicin stimulation of DRG neurons in culture facilitated the release of extracellular vesicles enriched with miR-23a, which were further on taken up in macrophages where, by interaction with the A20 gene, facilitated the macrophages’ switch towards a pro-inflammatory phenotype, as confirmed by increased Nos2 mRNA (M1 marker) and decreased Mrc1 mRNA (M2 marker) expression (Figure 6). A20 is a gene that encodes a ubiquitin-editing protein that is involved in the negative feedback regulation of NF-κB signaling. The effect was additionally confirmed by the fact that the intrathecal delivery of a miR-23a antagomir attenuated neuropathic hypersensitivity and reduced the number of M1 macrophages in injured DRGs [134].

The **miR-17-92 cluster**, a microRNA cluster with six distinct members, was shown to be upregulated in L5 rat DRG at 14 days after SNL, which was associated with significant **mechanical allodynia**, but not thermal hyperalgesia [135]. Through the overexpression of miR-17-92 members in L5 DRG using AAV vector injection, it was confirmed that of the six members of the cluster, only miR-18a, miR-19a, miR-19b or miR-92a were associated with mechanical allodynia. The upregulation of these miRNAs was associated with a downregulation of A-type K^+^ currents (Kv1.4, Kv3.4, and Kv4.3) only in small, nociceptive DRG neurons, while non-A-type K^+^ currents were not significantly affected (Figure 6). Since the suppression of K^+^ channels promotes increased neuronal excitability, the miR-17-92 cluster appears to perpetuate mechanical allodynia [135]. However, in this study, the macrophages’ contribution was not explored.

Increased levels of **miR-30b, miR-96, and miR-183** were correlated with increased expression in rat DRG neurons of different subtypes of Na_V_ channels and subsequent **increased pain sensitivity**: Na_V_1.7 after SNI [136], and Na_V_1.3 after CCI [137] and SNL [138], respectively. In contrast, a down-regulated level of **miR-182** was associated with the increased expression of Na_V_1.7 in L4-L6 rat DRGs from 3 days until 14 days after SNI [139] (Figure 6). The overexpression of miR-182 via microinjection reversed the abnormal increase of Na_V_1.7 at both the mRNA and protein level and significantly attenuated the mechanical hypersensitivity, while the administration of miR-182 antagomir enhanced the Na_V_1.7 expression at both the mRNA and protein level in L4-L6 DRG and, consequently, enhanced mechanical sensitivity in naive rats [139]. Again, in these studies, the macrophages’ contribution was not explored.

Several miRNAs were proved to have **anti-nociceptive** effects, alleviating neuropathic pain through different mechanisms (Figure 6). An upregulated level of **miR-122** was shown to be responsible for the analgesic effect of GCSF by suppressing CCL2 expression in rat DRG neurons 7 days after CCI [140]. **miR-449a** alleviates mouse SNI-induced pain by decreasing the level of calcium-activated K^+^ channel subunit α-1 (KCNMA1) and TRPA1 and by increasing the level of transmembrane phosphatase with tension homology (TPTE) in DRG neurons [141]. **miR-7a** ameliorates rat SNL-induced pain via the neurofilament light polypeptide-dependent signal transducer and activator of the transcription signaling pathway [142], and **miR-146a** ameliorates rat CCI-induced pain through the suppression of the IRAK1/TRAF6 signaling pathway in L4-L6 DRG [143].

For these miRNA molecules, which were proven as most promising candidates to facilitate either neuropathic pain development, but also pain resolution, a number of questions remain. What is the origin of these miRNAs, neurons or macrophages? Which is the initial trigger for their production and release: is it a mechanism inside the cell (like after a nerve injury) or an external pro-inflammatory mediator? Given their influence on neuronal excitability, what ionic channels are their final target, and which is the receptor that initiates the signaling pathway towards those putative ionic channels? Is there an injury-specific pattern of miRNAs? If yes, could it be that a specific miRNA is the most important, or more miRNAs are involved, in which case the network effect (some miRNAs increased; some miRNA decreased) is more important for the development/resolution of pain? Modulating the ion channels involved in increased neuronal excitability after injury would be more efficient if it would be made via miRNAs than by blocking other pro-inflammatory induced signaling pathways? These issues require further investigation.

There is still a debate out there on which macrophages are more important for the traumatic neuropathic pain development: those from the peripheral nerve or those from the sensory ganglia?

Studies on mice have shown that in a PSNL pain model, clodronate depletion of infiltrating monocytes in injured peripheral nerve significantly attenuated macrophage infiltration and allodynia [43,144], similar to an SNI pain model on MAFIA mice where chemogenic depletion of circulating monocytes and macrophages at the nerve injury site reduced the nerve-injury-induced mechanical hypersensitivity [145]. Even more, as these authors found that DRG macrophages were spared, they concluded that peripheral macrophages, but not those in the DRG, are the critical contributors to nerve-injury-induced neuropathic pain [145]. These data are in contrast with previous studies showing that selective depletion of peripheral monocytes/macrophages had limited impact on neuropathic pain development after spinal nerve transection in mice [146] or after SNL in rats [147], which raises the possibility of a type of injury-dependent mechanism for macrophage response at the peripheral nerve.

At the sensory ganglia level, in particular DRG, Yu et al. showed not only that in MAFIA mice resident macrophages from DRG seem to be more important than infiltrating ones in the development of SNI-induced neuropathic pain, but also that DRG macrophages and not the peripheral macrophages are critical for the initiation and persistence of SNI-induced neuropathic pain [42]. Similarly, clodronate-treated mice, which lost DRG macrophages after SNI, showed reduced neuropathic tactile allodynia and cold hypersensitivity [59]. The genetic analysis identified 36 transcripts whose expression closely correlated with the onset of mechanical sensitivity and were functionally related to immune function, and 137 transcripts closely correlated with the onset of cold sensitivity, which were functionally related to neuronal function, supporting the idea that resident macrophages in the DRG are crucial to the development and persistence of tactile, but not cold allodynia, at least after SNI.

## 7. Conclusions

The bidirectional communication established between activated macrophages clustered close to the traumatically lesioned peripheral nerves is very complex, involving both classic primary mediators, but also new miRNa-type primary mediators, released by both immune cells and neurons, with both pro- and anti-nociceptive effects. The fact that neurons are able to secrete substances similar to the ones released by macrophages in pathological conditions, illustrates how intricate the neuro-immune communication is. Since macrophages are the first sensors of defense activated after a lesion, understanding the dialogue they carry with neurons would allow the development of new therapeutic approaches for neuropathic pain, before peripheral sensitization or central sensitization develop.

## Figures and Tables

**Figure 1 ijms-23-12389-f001:**
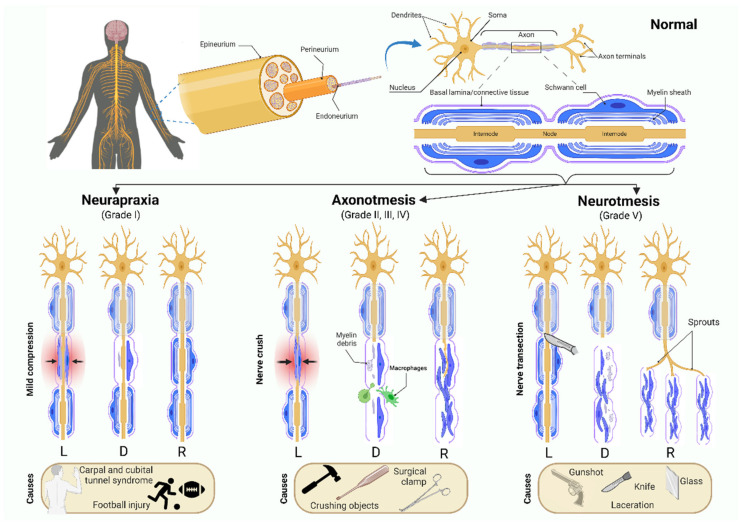
Different types of peripheral nerve lesions. Seddon (1943) classified them into neurapraxia, axonotmesis and neurotmesis, and Sunderland (1951) expanded on this classification to distinguish the extent of damage in the connective tissue: Grade I and Grade V correspond with Seddon’s neuropraxia and neurotmesis, respectively, while Grades II–IV are all forms of axonotmesis with increasing amounts of connective tissue damage. L—lesion; D—demyelination after neurapraxia or Wallerian degeneration after axonotmesis and neurotmesis; R—regeneration.

**Figure 2 ijms-23-12389-f002:**
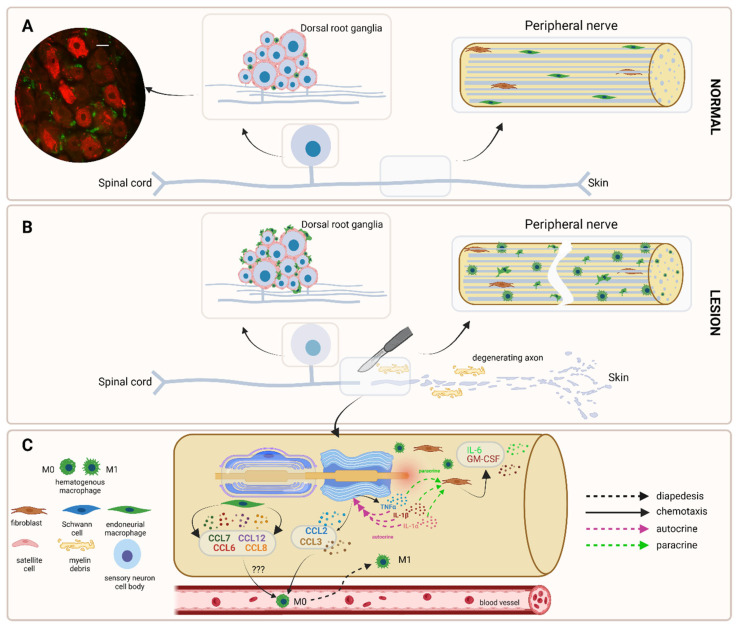
After a peripheral nerve lesion, macrophages are attracted to the lesion site. (**A**) Graphic representation of a normal DRG. Peripheral nerve consists of nerve fibers of different diameters and degrees of myelination, with elongated macrophages oriented along the longitudinal axis and scattered between nerve fibers. In normal conditions, few macrophages are scattered between DRG neurons as well. To the left, DRG immunostaining showing a small number of Iba1 (+) macrophages (green) scattered among large NF200 (+) neurons (red). Scale bar—20 μm. (**B**) Graphic representation of the peripheral nerve lesion site, where numerous endoneurial and hematogenous macrophages are attracted. (**C**) Signaling pathways that attract macrophages to the peripheral nerve lesion site. Lesioned Schwann cells release cytokines that in an autocrine manner induce the secretion of chemoattractant chemokines (CCL2, CCL3), which further on recruit hematogenous macrophages to the lesion site. Endoneurial macrophages, activated after the lesion, also release chemoattractant chemokines, which further increase the recruitment of hematogenous macrophages to the lesion site. IL-6 and GM-CSF secreted by activated fibroblasts do not directly participate in the recruitment of macrophages. *Legend*: DRG—dorsal root ganglia; Iba1—ionized calcium-binding adaptor molecule 1; NF200—neurofilament 200; CCL2, -3—chemokine (C-C motif) ligand 2, -3; IL-6—interleukin 6; GM-CSF—granulocyte-macrophage colony-stimulating factor.

**Figure 3 ijms-23-12389-f003:**
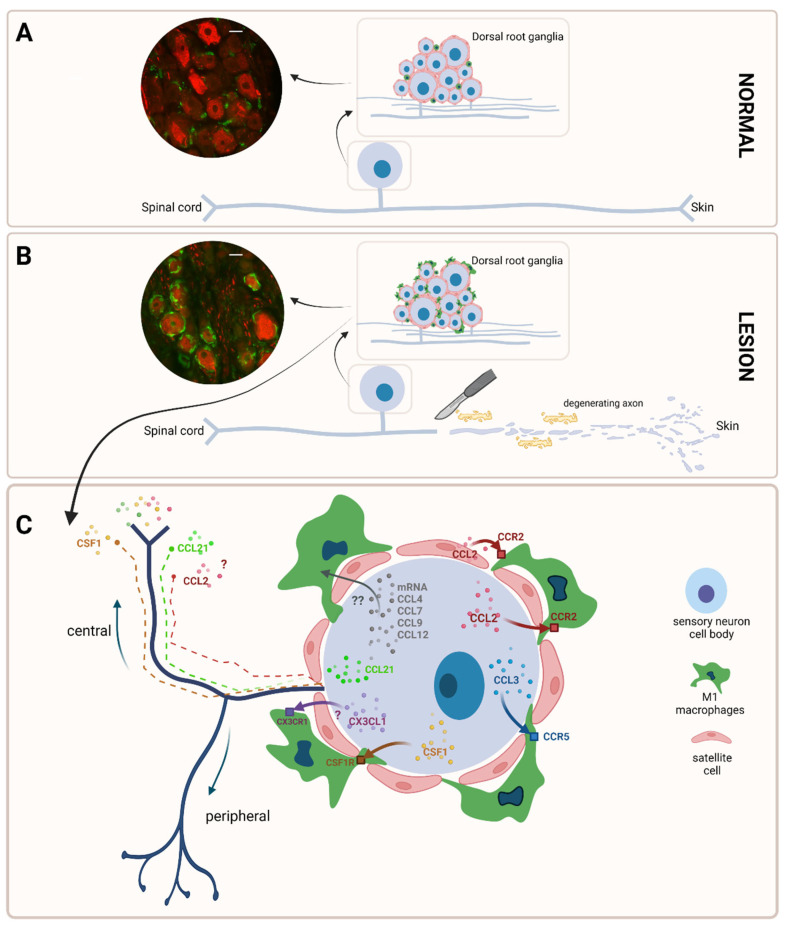
After a peripheral nerve lesion, macrophages are attracted around the DRG neurons. (**A**) Graphic representation of a normal DRG, with few macrophages scattered between DRG neurons. To the left, DRG immunostaining showing a small number of Iba1 (+) macrophages (green) scattered among large NF200 (+) neurons (red). Scale bar—20 μm. (**B**) Graphic representation of the DRG after the peripheral nerve lesion, with numerous endoneurial and hematogenous macrophages attracted around neurons. To the left, DRG immunostaining showing an increased number of Iba1 (+) macrophages (green) around large NF200 (+) neurons (red). Scale bar—20 μm. (**C**) Signaling pathways that attract macrophages around DRG neurons. CCL2, CSF1, and CCL3 are experimentally proven to contribute to clustering of macrophages around DRG neurons. For CX3CL1, CCL4, CCL7, CCL9, and CCL12, more experimental data are required to confirm their chemoattractant effects. CCL2, CSF1 and CCL21 may also act as secondary mediators at the spinal cord level. *Legend*: DRG—dorsal root ganglia; Iba1—ionized calcium-binding adaptor molecule 1; NF200—neurofilament 200; CCL2, -3, -4, -7, -9, -12, -21—chemokine (C-C motif) ligand 2, -3, -4, -7, -9, -12, -21; CX3CL1—chemokine (C-X3-C motif) ligand 1; CSF1—colony-stimulating factor 1.

**Figure 4 ijms-23-12389-f004:**
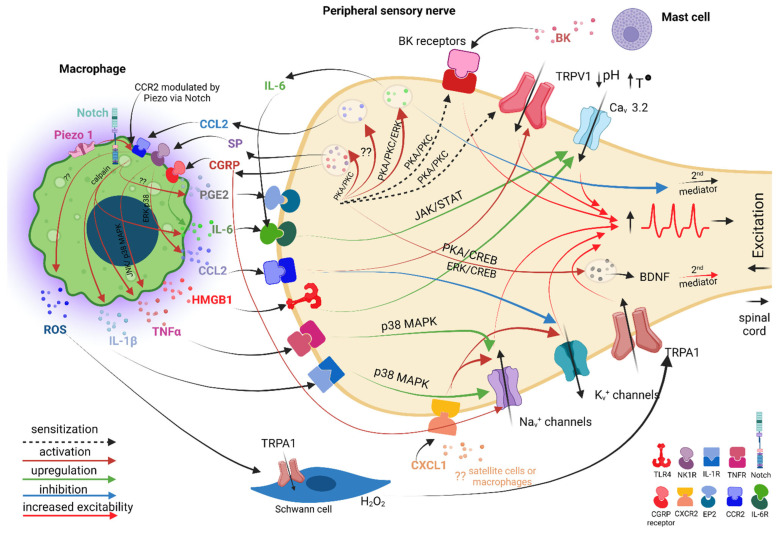
Illustration of the pro-nociceptive dialogue between macrophages and peripheral sensory neurons after traumatic peripheral nerve injury. Macrophages attracted in the vicinity of lesioned neurons switch to M1 phenotype and initiate with the neurons a pro-nociceptive dialogue mediated by specific primary mediators, which increases neuronal excitability. Some of the primary mediators released by macrophages act directly on neurons via specific receptors (i.e., PGE2, IL-6, CCL2, HMGB1, TNF-α, IL-1β) or indirectly via Schwann cells (i.e., ROS) to upregulate/activate ionic channels (i.e., Na_v_ channels, TRPA1, TRPV1, Ca_v_3.2 channels), downregulate ionic channels (i.e., K_v_ channels), or sensitize specific receptors (BK, TRPV1), with the final end results of more action potentials generated. These action potentials will propagate to the brain via the spinal cord, where they will be interpreted as pain. In response, lesioned primary sensory neurons will release other primary mediators, which in a positive feedback loop will act on macrophages to further activate them (i.e., SP, CGRP, CCL2), or will act in an autocrine manner to activate neurons (i.e., IL-6, CGRP). Some of the substances secreted by the lesioned neurons as a response to communication with the macrophages may also act as secondary mediators, further up at the spinal cord, where they can either facilitate the activation of microglia (= more pain) (i.e., BDNF) or desensitize microglia (= less pain) (i.e., IL-6). *Legend*: PGE2—prostaglandin E2; IL-6—interleukin 6; CCL2—chemokine (C-C motif) ligand 2; HMGB1—high-mobility group box 1; TNF-α—tumor necrosis factor alpha; IL-1β—interleukin 1 beta; ROS—reactive oxygen species; TRPA1—transient receptor potential ankyrin 1; TRPV1—transient receptor potential vanilloid type-1; BK—bradykinin; SP—substance P; CGRP—calcitonin gene related peptide; BDNF—brain derived neurotrophic factor.

**Figure 5 ijms-23-12389-f005:**
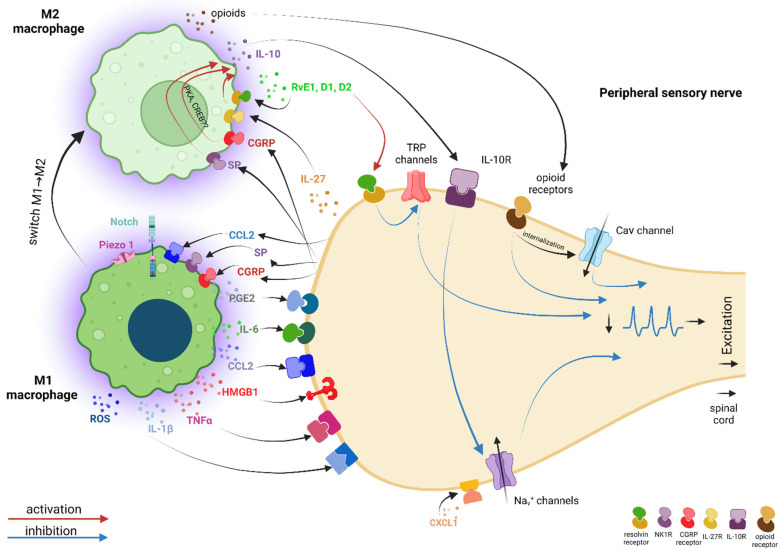
Illustration of the anti-nociceptive dialogue between macrophages and peripheral sensory neurons after traumatic peripheral nerve injury. Some of the primary mediators released by lesioned neurons as part of the dialogue with activated M1 macrophages not only further activate macrophages, but also initiate the secretion of anti-inflammatory cytokine IL-10 specific to the M2 anti-inflammatory phenotype of macrophages (i.e., SP, CGRP). In addition, lesioned neurons may also secrete IL-27, which also stimulates the IL-10 release by macrophages, further augmenting their switch to the M2 phenotype. IL-10 acts back on neurons, where it inhibits Na_v_ channels, decreasing their excitability and reducing pain. In the M2 state, macrophages additionally release opioids (met-enkephalin, dynorphin and β-endorphin), which act back on neurons where, via specific opioid receptors, they may induce the internalization of Ca_v_ channels which decreases neuronal excitability and reduces pain. Additionally, M2 macrophages may secrete resolvins (E1, D1 or D2), which back on neurons may inhibit TRP channels via GPCR receptors, or may promote the M2 phenotype in an autocrine manner. *Legend*: SP—substance P; CGRP—calcitonin gene-related peptide; IL-10—interleukin 10; IL-27—interleukin 10; Rv E1, D1, D2—resolvins E1, D1 and D2; GPCR—G-protein-coupled receptors.

**Figure 6 ijms-23-12389-f006:**
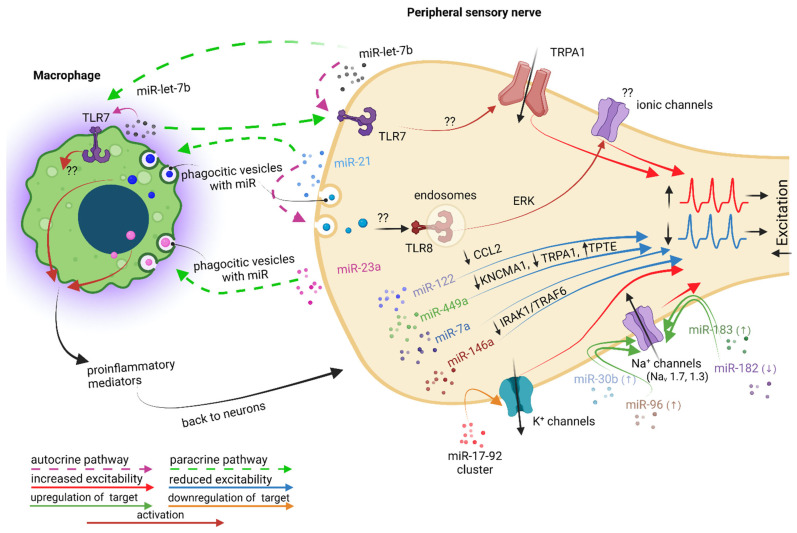
Illustration of the pro- and anti-nociceptive dialogue between macrophages and peripheral sensory neurons mediated (or putatively mediated) by miRNAs after traumatic peripheral nerve injury. Traumatically lesioned neurons release miRNAs that have pro-nociceptive effects, exerted via macrophages or in an autocrine manner, back on neurons. miR-21 and miR-23 released by lesioned neurons are engulfed my neighboring macrophages, at the level of which promote the M1 switch and downstream secretion of pro-inflammatory primary mediators, which act back on neurons to increase their excitability. The miR-17-92 cluster and miRs-39b, -96 and -183 released by lesioned neurons seem to act in an autocrine manner on neurons, where they induce increased excitability via the downregulation of K_v_ ion channels or the upregulation of Na_v_ ion channels, respectively. A similar effect on Na_v_ ion channels is induced by a reduced expression of miR-182. miR-let-7b is the only one proved to be secreted both by neurons and macrophages, but its role in traumatic induced neuro-immune communication is putative, since there are not enough experimental data. Lesioned neurons also secrete anti-nociceptive miRNAs (i.e., miR-122, miR-449a, miR-7a and miR-146a) for which the analgesic effects were proved together with the signaling pathways they activate, although the final molecular target is not yet clear.

## Data Availability

Not applicable.

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
