# Peer review of "The Yin/Yang Balance of Communication between Sensory Neurons and Macrophages in Traumatic Peripheral Neuropathic Pain"

_ijms, 2022, doi:10.3390/ijms232012389_

Round 1

Reviewer 1 Report

This review systematically present that ion channels as facilitators of sensory-neurons-macrophages communication in traumatic peripheral neuropathic pain. As a systematic review, this article has certain practical value and is helpful to further studies in traumatic neuropathic pain.

Here are some of my opinions.

1. The review is entitled Ion channels as facilitators of sensory-neurons-macrophages communication in traumatic peripheral neuropathic pain”. However, most of the content was about macrophages and signaling pathways while very little refer to ion channels.

2. In the abstract, the authors wrote that the final aim of this review is to understand what is the role of ion channels, in both neurons and macrophages, as facilitators of this communication. However, the text mainly discussed the unique characteristics of nerve associated macrophages in the peripheral nerves and sensory ganglia and of the molecules and signaling pathways. It would be more comprehensive and explicit to elaborate from the above aspects.

3. The ion channels involved in traumatic peripheral neuropathic pain should also be introduced comprehensively. Some relevant articles can be cited. I suggest that authors refine more of their own views through the existing literature.

Reviewer 2 Report

The aim of this review is an overview of the unique characteristics of nerve associated macrophages in the peripheral nerves and sensory ganglia and of signaling pathways involved in the neuro-immune cross-talk after a traumatic lesion, with the final aim to understand what is the role of ion channels, in both neurons and macrophages, as facilitators of this communication.

In general, the paper is well written, well organized to address the aims of the review. The figures are informative and summarizing the narrative.

Specifically:

Page 4, 1st and 2nd para. In the first para the authors indicate the differences between nerves of PNS in terms of gene expression. Next, in the conclusion, the authors indicate that macrophages in sensory nerves and DRG ganglia are similar to macrophages in sympathetic ganglia, which are not sensory ganglia. The relevance of this comparison is not clear.

Page 5 Ln 212.”  “SNL” and “sciatic nerve ligation” is it not the same?

Page 13. Describing the “anti-nociceptive” role of macrophages, is worth mentioning of resolvn’s production by macrophages, that have been shown play role in resolution of inflammation and pain (PMID: 20383154).
